# *Sargassum* accumulation and transport by mesoscale eddies

Rosmery Sosa-Gutierrez[1,2], Julien Jouanno[1], Leo Berline[3]

1 LEGOS, Université de Toulouse, IRD, CNRS, CNES, UPS, Toulouse, France

CELAD/Mercator Ocean International, Toulouse, 31400, France

3Aix-Marseille University, Université de Toulon, CNRS/INSU, IRD, MIO UM 110, Mediterranean Institute of Oceanography (MIO), Campus of Luminy, 13288 Marseille, France

*Correspondence to*: Rosmery Sosa-Gutierrez (rsosa@mercator-ocean.fr), Julien Jouanno (julien.jouanno@ird.fr)

**Abstract.** The proliferation of pelagic *Sargassum* spp. (Sargassum) in the tropical Atlantic has significant ecological and socio-economic impacts. While large-scale ocean circulation patterns influence the basin scale distribution of *Sargassum*, the
role of mesoscale eddies in their local accumulation and transport has not been quantitatively assessed so far. This study investigates the relationship between mesoscale eddies and *Sargassum* dynamics using satellite observations. By analyzing 13 years of remote sensing observations, we demonstrate that both cyclonic and anticyclonic long-lived mesoscale eddies can trap and transport *Sargassum*. However, results show that in cyclonic eddies *Sargassum* cover is higher and tends to accumulate during their lifetime while within anticyclonic eddies the Sargassum cover is usually weaker and tend to decrease. These
findings align with recent studies highlighting the role of eddies in shaping the distribution of floating debris and provide an important observational basis for the development of *Sargassum* drift models.

## 1 Introduction

Prior to 2010, the two holopelagic *Sargassum* species, *Sargassum natans* and *Sargassum fluitans* (*Sargassum* hereafter), were primarily found in the Sargasso Sea and the northwestern tropical Atlantic (Gower and King 2008). However, since 2010,
these two pelagic *Sargassum* species have expanded their presence to the Tropical Atlantic (< 20° N), from the coasts of the Lesser Antilles, Central America, Brazil, to West Africa (Gower et al. 2013, Wang et al., 2019). This expansion has led to stranding events and large accumulations, resulting in significant economic and environmental damage on the coast areas of the tropical Atlantic (e.g., Rodríguez-Martínez et al. 2024, Van Tussenbroek et al. 2017, Hendy et al., 2021, Antonio-Martínez et al., 2020, Rosellón-Druker et al. 2023).

Sargassum remains afloat in the upper ocean due to its gas-filled bladders, making it highly responsive to both surface currents and wind. The dynamics of the upper ocean play a critical role in the formation of *Sargassum* accumulations, which can occur across a wide range of spatial scales (see Ody et al., 2019). At smaller scales, on the order of tens of meters, accumulations are typically driven by Langmuir circulation (Langmuir, 1938). At larger scales, reaching hundreds of kilometers, mesoscale and submesoscale frontal dynamics become dominant (Gower et al., 2013; Zhong et al., 2012). In particular, convergence zones
associated with submesoscale dynamics have been shown to concentrate buoyant material and to structure ecosystems (D'Asaro et al., 2018; Esposito et al., 2021, Lévy et al. 2018). However, the role of mesoscale eddies in the accumulation and transport of *Sargassum* remains uncertain. Early theoretical and experimental work by Provenzale (1999) suggested that heavy impurities can be concentrated in the cores of anticyclonic vortices. Beron-Vera et al. (2015) provided both theoretical justification and numerical evidence for a more general principle governing the behavior of inertial particles near

quasigeostrophic eddies: anticyclonic (cyclonic) eddies tend to attract heavy (light) particles and repel light (heavy) ones, respectively. More recent advances, incorporating wind drag and elastic forces into the Maxey–Riley equations, have shown that these additional factors can have opposing influences sometimes favoring accumulation in anticyclones rather than cyclones (Beron-Vera, 2021). Observational studies also present a mixed picture. For instance, limited in situ measurements have shown microplastic accumulation within anticyclonic eddies (Brach et al., 2018), while a more systematic study by Vic et al. (2022) demonstrated a tendency for drogued drifters to accumulate in cyclonic structures in the North Atlantic. These contrasting findings highlight the complexity and context-dependence of floating object dynamics in mesoscale eddies. The accumulation and transport behavior likely depend on the specific properties of the objects in question—especially their buoyancy and windage.

With respect to *Sargassum*, a few case studies have shown that mesoscale eddies can transport *Sargassum*. For example, Andrade-Canto et al. (2022), using 8 years of satellite altimetry in the eastern Caribbean Sea, showed that mesoscale eddies (both cyclonic and anticyclonic) can transport *Sargassum*, and more recently Sun et al. (2024) presents some illustrative cases of accumulation in eddies. However, there has been no systematic assessment of the transport and organization of *Sargassum* by these eddies. We fill this gap through a systematic analysis of *Sargassum* distribution by combining eddy tracking techniques from altimetry (Chaigneau et al. 2009, Pegliasco et al. 2015, Sosa-Gutierrez et al. 2020) and long-term *Sargassum* detection from MODIS ocean color sensor (Descloitres et al. 2021) for the last 13 years. There are several known eddy detection and tracking algorithms based on different methods, such as Okubo-Weiss (Okubo 1970; Weiss 1991), vector-geometry (Nencioli et al. 2010), winding angle (Chaigneau et al. 2009; Chen et al. 2011), and recently, geodesic detection (Andrade-Canto et al., 2022). In this work, we opted for an Eulerian detection method due to its relative ease of implementation. This paper is organized as follows. Section 2 describes the data and methods used. The main characteristics of *Sargassum* cover in cyclonic and anticyclonic eddies are presented in the results section, section 3. Finally, in section 4, we present the summary and discussion of our results.

## 2 *Sargassum* and eddy detection methods

*Sargassum* detections were obtained from the SAREDA database (*Sargassum* Evolving Distributions in the Atlantic, Descloitres et al., 2021). This product estimates *Sargassum* cover by computing the Alternative Floating Algae Index (AFAI; Wang and Hu 2016) from ocean color acquisitions by the Moderate Resolution Imaging Spectroradiometer (MODIS), which operates aboard the Aqua and Terra satellites. The AFAI, computed using the processing described in Descloitres et al. (2021), was converted to Fractional Cover (FC), which represents the proportion of *Sargassum* cover in each pixel. Daily FC at 1 km from the SAREDA database were aggregated on a regular grid of 0.25° (~25 km) horizontal resolution.These daily MODIS images have significant cloud coverage compared to multi-sensor products (e.g., Sun et al., 2024) but have the advantage of being a homogeneous series over the last 13 years. They have already allowed the tracking of rapid decrease in *Sargassum* coverage in the lee of tropical cyclones (Sosa-Gutierrez et al. 2022). They therefore seem well suited to the mesoscale compositing proposed in the present study. *Sargassum* biomass is obtained from *Sargassum* cover by considering an average density of 3.34 kg m$^{-2}$ for pure *Sargassum* patches as estimated by Wang et al. (2018).

Mesoscale cyclonic and anticyclonic eddies in the Tropical Atlantic were identified from 2011 to 2023 using daily Absolute Dynamic Topography (ADT) maps at a horizontal resolution of 25 km, which are distributed by the Copernicus Marine and Environment Monitoring Service (CMEMS; https://www.copernicus.eu/en). The mesoscale eddies were detected using a method described by Chaigneau et al. (2009), with a slight modification of the methodology outlined by Sosa-Gutierrez et al.

(2020). While the original method identifies the outermost closed contour of Sea Level Anomalies (SLA) for each anticyclonic and cyclonic eddy center (SLA maxima and minima, respectively) as eddy edges, the modification involves considering the outermost closed SLA where the averaged azimuthal velocity is maximum at the periphery of the eddy. The eddy-tracking algorithm, described in Pegliasco et al. (2015), involves calculating the paths of eddies by intersecting them with daily maps. For this study, only mesoscale eddies with radii larger than 40 km and lifetime greater than 30 days were retained.

An illustration of 7-day averaged *Sargassum* cover centered on June 28, 2021, together with superimposed eddy detections for the same day, is given in Figure 1. This figure illustrates that the *Sargassum* distribution is partially structured by mesoscale eddies. To provide a systematic and quantitative view of this organization by mesoscale eddies, we composited the *Sargassum* coverage by selecting eddies under the following conditions: i) the detected eddy last at least 30 days, ii) during its lifetime the eddy has more than 25% of unclouded ocean color observations, iii) *Sargassum* presence is detected at least once during the eddy lifetime. This results in 471 trajectories of anticyclonic eddies (AE) and 628 trajectories of cyclonic eddies (CE) between 2011 and 2023 (Figure 2a and b, respectively). The trajectory analysis shows that AEs are predominantly found in the path of the western boundary current from the North Brazil Current (NBC) to the Loop Current, in agreement with known distribution of large anticyclonic eddies in this region (e.g. Richardson 2005, Jouanno et al. 2008). In contrast, CEs are more homogenously distributed in the central Atlantic and Caribbean Sea. The mean lifetime of AEs that captured *Sargassum* was $69 \pm 40$ days (mean $\pm$ standard deviation) and $65 \pm 37$ for CE; the longest-lived AE lasted 277 days and 269 days for a CE, respectively (Figure 2c-d).

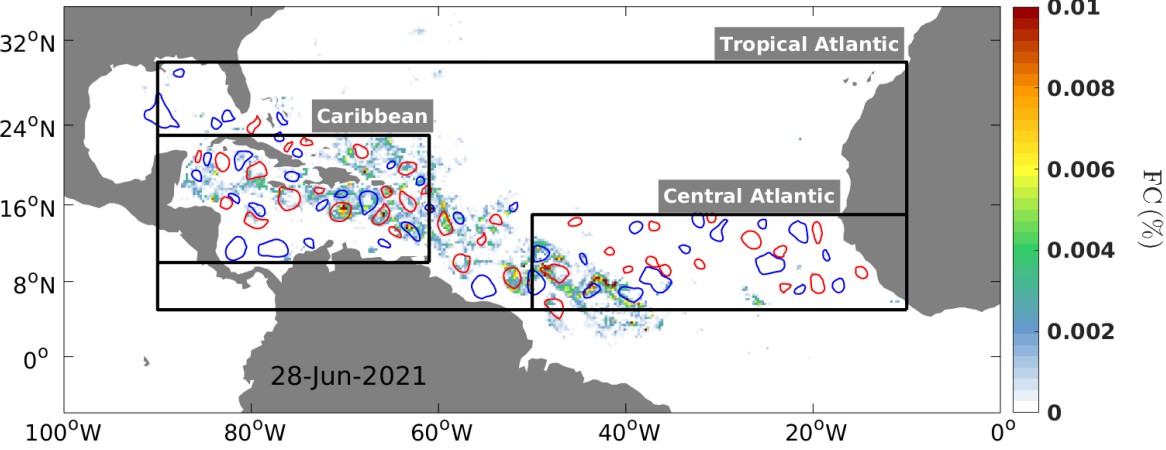

**Figure 1: Distribution of *Sargassum* cover (%) obtained from MODIS and mesoscale eddy contours detected with CMEMS ADT product for day June 28, 2021. *Sargassum* biomass was averaged over a 7-day period centered on day June 28 and coarsen to 0.25° regular grid. Red contours represent anticyclonic eddies, and blue contours represent cyclonic eddies. Only eddies with non-zero FC during their lifetime are shown.**

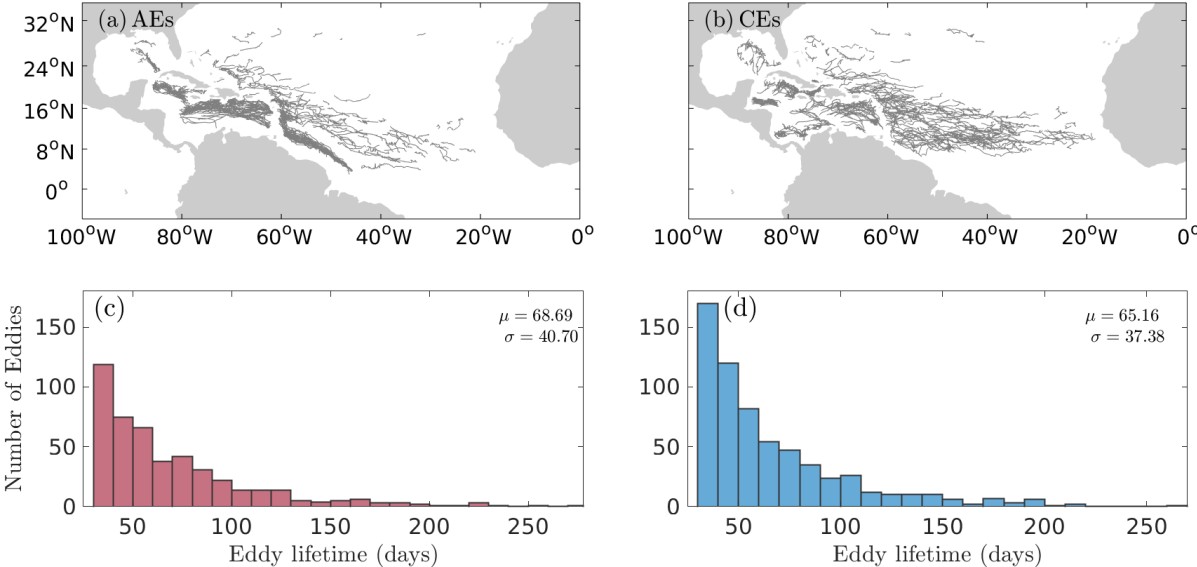

**Figure 2: Eddy tracks for the period 2011-2023 with (a) trajectories of AEs and (b) trajectories of CEs that last at least 30 days and have seen *Sargassum* during their lifetime. (c) and (d) show histograms of the total number of AEs (red) and CEs (blue) with respect to eddy lifetime (in days).**

The *Sargassum* cover within CEs and AEs and their neighborhood was averaged for each day within the period 2011-2023 for the entire Tropical Atlantic (10°W-90°W, 5°N-30°N) and in two regions where mesoscale activity and *Sargassum* presence are significant: the Caribbean Sea (61°W-90°W, 10°N-23°N), and the Central Atlantic (10°W 50°W, 5°N-15°N). To construct the composite distribution, each eddy was normalized by its radius, where radius of ±1 corresponds to the eddy periphery, 0 corresponds to the eddy center, and radius larger than +1 or lower than -1 corresponds to the region surrounding the eddy (Figure 3). This normalization allowed to account for eddies of different sizes. Since the composites are constructed using eddies where *Sargassum* presence is detected at least once during the eddy's lifetime, this could introduce a bias favoring instances of *Sargassum* being trapped inside eddies. To verify that, we performed a null hypothesis test by compositing the *Sargassum* distribution with the same criteria but using the eddy contours for the previous year. The results are shown in Figure 4 and confirm that the eddy trapping observed in Figure 3 and discussed below is not a bias of the methodology.

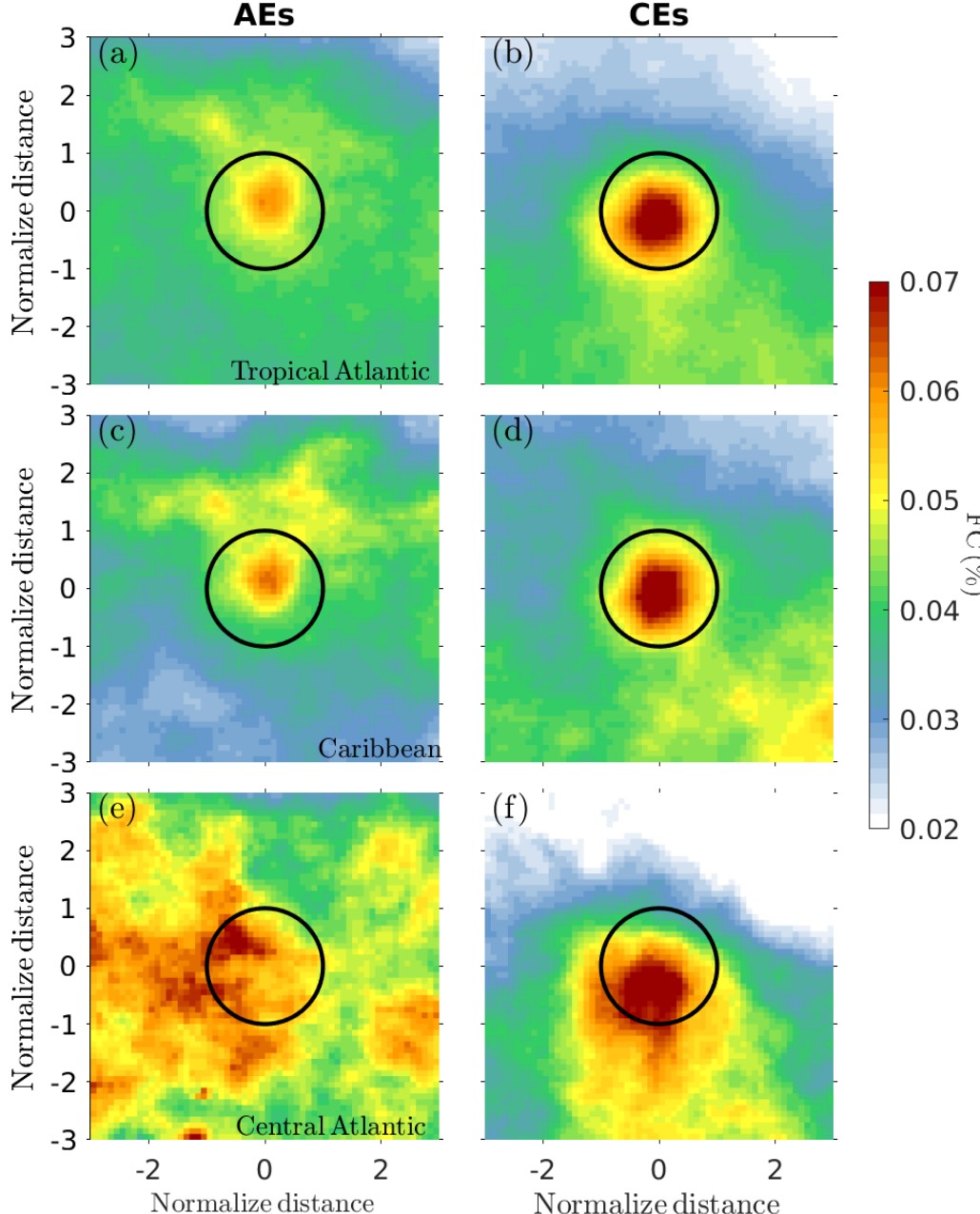

Figure 3: Normalized composite averages of *Sargassum* cover (%; color) for AEs and CEs, for the following regions Tropical Atlantic (top), Caribbean (middle), and Central Atlantic (bottom). The black circle represents an idealized contour of the eddy periphery.

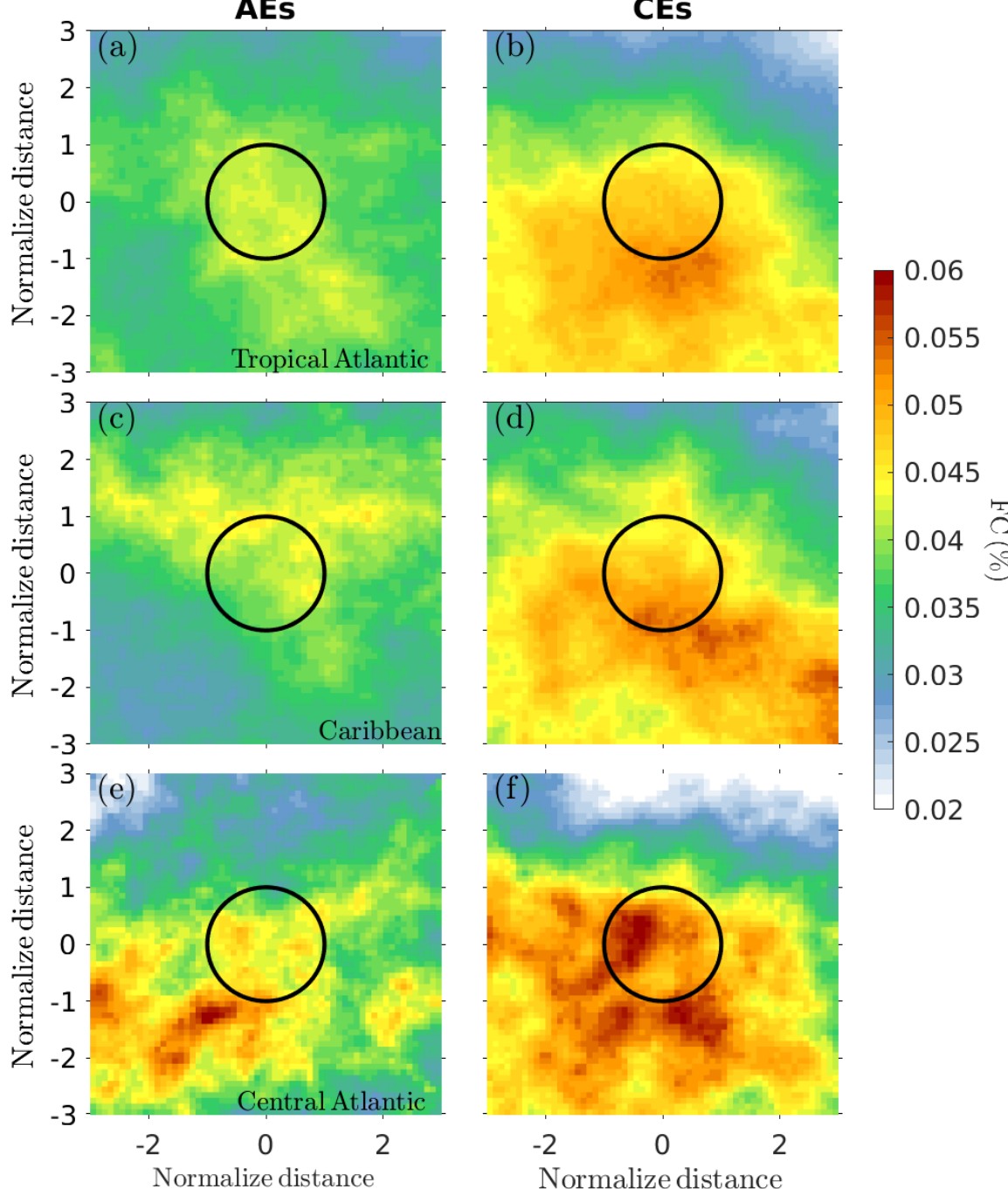

**Figure 4. Same as Figure 3, but the *Sargassum* cover observed for a given day have been composited using eddy detections from the previous year. The same selection criteria were used as in Figure 3. This is a null hypothesis that allows to verify that the selection of the eddy cases for the compositing (e.g. that only eddies where *Sargassum* was observed are considered) does not induce a bias in favor of *Sargassum* accumulation inside the eddies.**


## 3 Results

The *Sargassum* cover composited for AEs and CEs in the Tropical Atlantic is shown in Figure 3ab. It reveals that both anticyclones and cyclones accumulate *Sargassum* in their core, but with much greater accumulation of *Sargassum* in cyclones (Figure 3b) than in anticyclones (Figure 3a). There are regional differences in this distribution. In the Caribbean, for example, the contrast between AEs and CEs is pronounced (Figure 3c-d and 5d), whereas in the central Atlantic (Figure 3e-f and 5g) it is much less pronounced with much more spread distribution for CEs, and no evidence for accumulation in AEs. This could

be due to the small number of AEs samples in this region, Table 1. Also, note in Figure 3b a meridional offset between the location of the maximum FC peak and the center of the eddies unlike the Caribbean composites (Figure 3d), where the *Sargassum* distribution appears centered within the eddies. The pattern observed in Figure 3b may result from the influence of Central Atlantic cases included in the composite, which show a southward displacement of *Sargassum* relative to the eddy centers (see Figure 3f). The cause of this southward shift is not fully understood at this stage, but it may be linked to windage

effects from the trade winds or to the background distribution of *Sargassum*, which generally tends to accumulate along the Intertropical Convergence Zone (ITCZ). Figure 4, also shows this latitudinal gradient of FC (see Figure 4b and 4d), which means that this gradient is a result of sampling this peculiar region with this eddy scale.

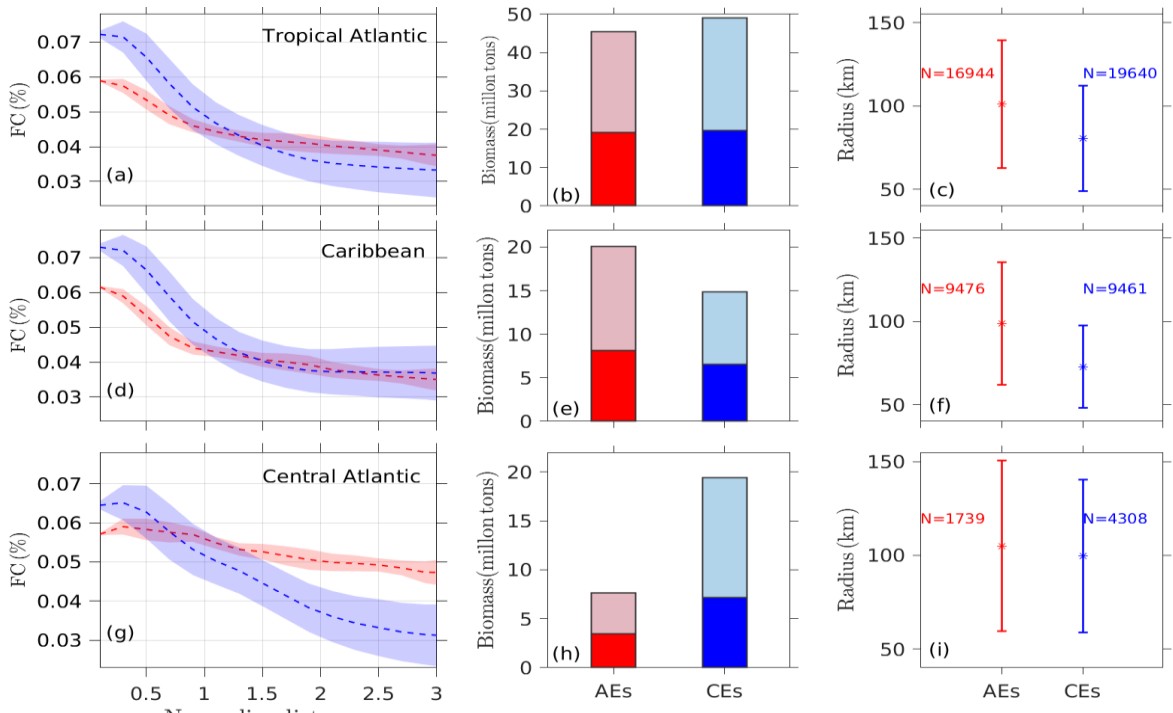

**Figure 5: (a, d, g)** *Sargassum* **cover averaged within concentric radii (represented as normalize distance), the innermost radius was**
**R = 0.1, with an interval of 0.2, until reaching radius =3 (red and blue lines correspond to AE and CE, respectively), and the red and blue shading represent one standard deviation to AE and CE. (b, e, h)** *Sargassum* **biomass within the AEs and CEs (light red and blue bars) compared with the** *Sargassum* **biomass in eddies propagating for at least 30 days (dark red and blue bars). (c, f, i) Radius**


distribution (±1 standard deviation) of all eddies that contributed to build the composite, where N represents the number of eddies used for the calculation of the composites.

In terms of biomass, the AEs and CEs transport similar amount of *Sargassum* at the basin scale (Figure 5b) but again with significant regional differences (Figure 5e-h). In the Caribbean, despite greater accumulation of *Sargassum* by cyclonic eddies, the anticyclones contain more *Sargassum* on average. This is explained by the difference in size between the two types of eddies in this region, with anticyclones being much larger than cyclones (Figure 5f). In the central Atlantic, there are many

more cyclonic eddies (Figure 5i), which favors a higher biomass in cyclonic eddies. To quantify these differences in Table 1 we show the ratio of total area and biomass ratio of CEs vs AEs. In the Caribbean, while the area ratio CEs/AEs is of 0.54, the accumulation of *Sargassum* in CEs lead to a biomass ratio CEs/AEs of 0.80. These regional differences are well illustrated in Figure 6ab, which shows a large *Sargassum* content in the AEs of the North Brazil Current, the NBC rings pathways, and the north Caribbean. The high *Sargassum* content transported by cyclonic eddies around 10°N corresponds to the shear zone

between the North Equatorial Current and the North Equatorial Countercurrent (Figure 6ab).

| Region | Type of eddies | Mean radius (km) | Numbers of eddies (N) | Mean biomass (million tons) | Area ratio (CEs/AEs ) | Biomass ratio (CEs/AEs) |
|---|---|---|---|---|---|---|
| **Caribbean** | CEs | 72.7 | 9461 | 6.52 | 0.54 | 0.80 |
| | AEs | 98.7 | 9476 | 8.14 | | |
| **Central Atlantic** | CEs | 99.77 | 4308 | 7.13 | 2.23 | 2.06 |
| | AEs | 105 | 1739 | 3.45 | | |

Table 1.Statistics of *Sargassum* distribution within eddies in the Caribbean Sea and Central Atlantic.

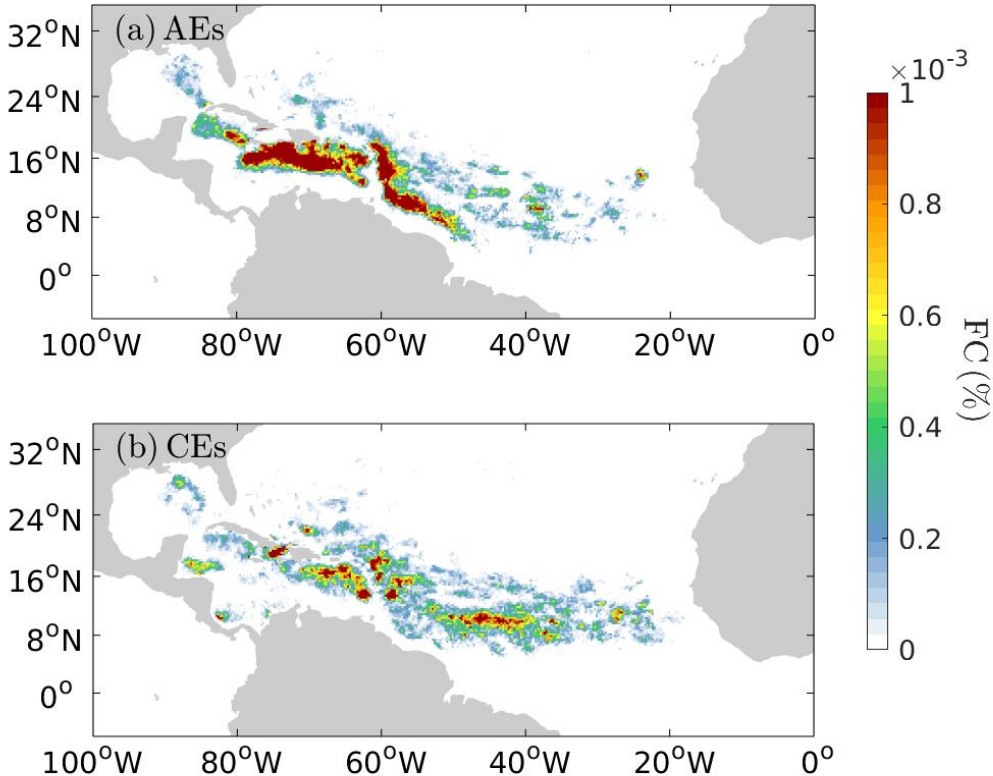

**Figure 6. The *Sargassum* cover (%) within AEs and CEs during the period from 2011 to 2023.**

The distribution of the *Sargassum* varies along the eddy life cycle as revealed by the evolution of the *Sargassum* cover with
respect to the normalized lifetime of the eddies (Figure 7). It shows that CEs increase their *Sargassum* cover along time for all
the region considered, while the *Sargassum* cover within AEs show a tendency to decrease along time, especially for the
Caribbean Sea. The increase of the *Sargassum* cover at the end of the life of the CEs in the central Atlantic could be explained
by Ekman pumping. This mechanism depends on wind stress (trade winds across this region) and eddy velocity (Gaube et al.,
2014), allowing convergence (divergence) of Ekman transport in CEs (AEs). This mechanism could favor the concentration
of *Sargassum* in CEs. It is worth noting that the occurrence of AEs in the Central Atlantic is low (Figure 2) and therefore may
contribute to a much noisier signal than in the Caribbean (Figure 7b).

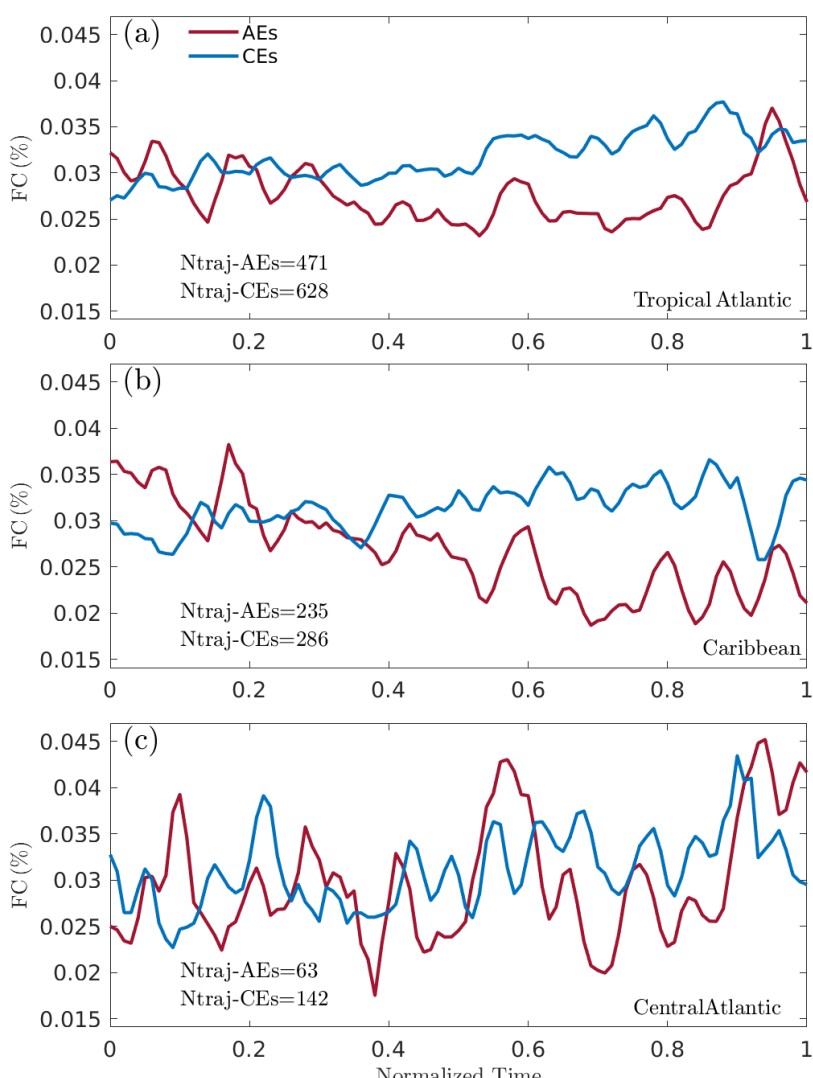

**Figure 7: (a) Average temporal evolution of *Sargassum* cover as a function of the normalized eddy lifetime (AEs and CEs, indicated by red and blue lines, respectively). Ntraj-AEs and Ntraj-CEs are the number of trajectories considered for the different regions.**

Figure 8a shows the mean monthly biomass of *Sargassum* aggregated across the tropical Atlantic (5-30°N and 0-100°W), showing a clear seasonal cycle characterized by a growth phase from January to July, followed by a decline phase from September to December. In addition, estimates of *Sargassum* biomass within the AEs and CEs are shown (Figure 8a). This provides insight into the periods when the retention of *Sargassum* biomass within the eddies was significant. Figure 8b illustrates the monthly percentage of *Sargassum* biomass contained within AEs and CEs relative to the total biomass. The accumulation of *Sargassum* biomass by these eddies can be significant, reaching up to 20% in certain months. For example, in June 2018, this amount was equivalent to an estimated biomass of approximately 3 million tons (Mt) of *Sargassum*.

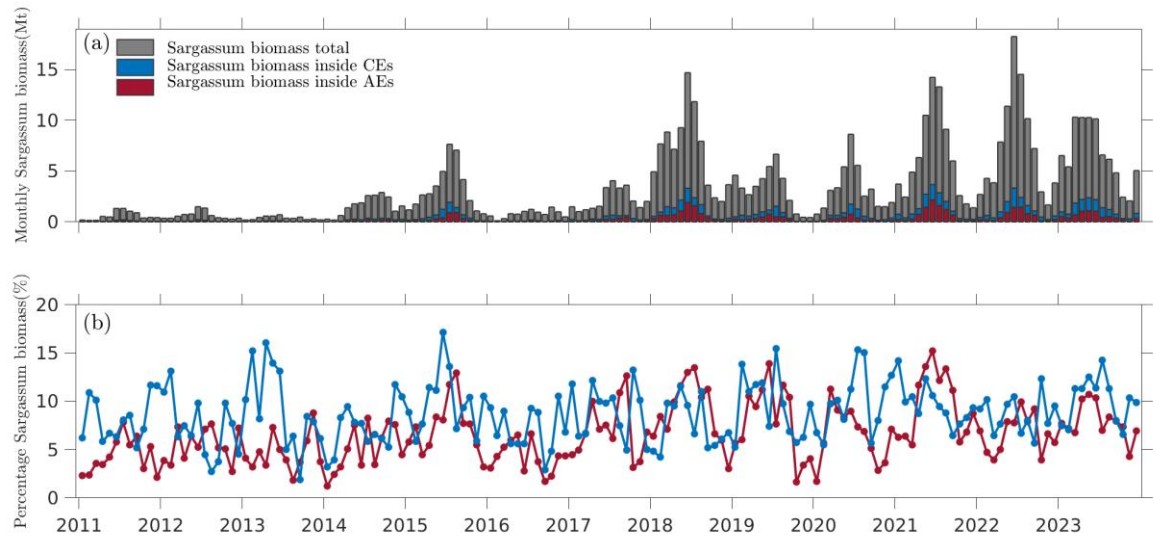

Figure 8: (a) The total monthly biomass of *Sargassum* in the tropical Atlantic (gray bars, 5-30°N and 0-100°W), and within the AEs and CEs (red and blue bars in stacked form, respectively). (b) The monthly percentage of *Sargassum* biomass retained within the AEs and CEs relative to the total biomass (indicated by red and blue lines, respectively).

*4 Conclusion and Discussion*

This research provides new insights into the important role played by oceanic mesoscale eddies in structuring *Sargassum* biomass in the tropical Atlantic and fills a gap in our understanding of *Sargassum* organization by the ocean dynamical continuum, between the fine scale aggregation processes (Langmuir 1938, Zhong et al. 2012) and the large-scale distribution (Wang et al. 2019, Jouanno et al. 2023).

Using ADT satellite imagery, the distribution of *Sargassum* detections from 2011 to 2023 was analyzed, by applying an eddy detection and tracking methodology (Chaigneau et al., 2009; Pegliasco et al., 2015). By combining these observational datasets, we have shown that both mesoscale cyclonic and anticyclonic mesoscale eddies can trap and transport *Sargassum*. This suggests that these eddies serve as effective *Sargassum* transporters. The eddy detection method used in this study (Chaigneau et al., 2009) has been widely applied in the literature (e.g., Pegliasco et al., 2021; Ernst et al., 2023). Like any method, however, it has certain limitations. Its performance can be influenced by the resolution of the satellite data and the specific detection criteria employed. It is also important to exercise caution when interpreting Eulerian mesoscale eddy detections, as these rely on streamline-based approaches that are inherently observer-dependent (Andrade-Canto and Beron-Vera, 2022). In contrast, Lagrangian methods, such as geodesic eddy detection, identify eddies as coherent material structures that resist stretching and deformation over time, and are therefore better suited to capturing flow-invariant transport pathways. In this work, we opted for an Eulerian detection method due to its relative ease of implementation. While it would be possible to reproduce this analysis using the methodology of Andrade-Canto and Beron-Vera (2022), we do not expect fundamentally different results, particularly because our detection method does not identify eddies that cross the Antilles Arc or the Yucatan Channel (see Figure 2), as was the case in Huang et al. (2021). That said, Lagrangian approaches would be particularly valuable for investigating transport and accumulation mechanisms, especially in scenarios where eddies re-form in the wake of islands.

Our composite analysis of *Sargassum* cover within mesoscale eddies consistently shows a preference for *Sargassum* accumulation within CEs over AEs in the tropical Atlantic, with on average 15% higher *Sargassum* cover in CEs. Although using a drogued drifter dataset, this is consistent with recent findings by Vic et al. (2022), who show a 24% higher accumulation of drifters in CEs compared to AEs in the North Atlantic using observed and simulated trajectories. The tendency to accumulate or not in cyclonic and anticyclonic eddies has been shown to depend on theoretical choices and complexity of drift models (Provenzale, 1999, Beron-Vera, 2021), so our findings should have important implications for motivating future drift model developments.

Results also suggest that the trapping is much more effective in the Caribbean than in the Central Atlantic, as revealed by sharpest contrast of *Sargassum* cover between the interior and exterior in the Caribbean. One hypothesis is that there is a much less energetic mesoscale activity in the Central Atlantic compared to the Caribbean: weak eddy circulation and the drift associated with the trade winds may reduce the capacity of the eddies to accumulate *Sargassum*.

These results raise the question of how mesoscale activity could influence the growth and decay of *Sargassum* by modulating nutrient availability. It has been documented that processes such as Ekman pumping and eddy pumping act by increasing nutrients in the euphotic layer in cyclones and decreasing nutrients in the euphotic layer in anticyclones (Gaube et al., 2014, McGillicuddy, 2016). In addition to contrasted surface radial velocity properties between AEs and CEs, this could contribute to the contrasted time evolution of the distribution highlighted in Figure 7 and could modulate overall growth of total *Sargassum* biomass at the basin scale. This should receive further attention in the future.

*Code and data availability.* All code and processed data needed to reproduce the main results and figures in this paper have been made available via Zenodo (https://zenodo.org/records/14816717)

*Author contributions.* RSG, JJ and LB conceived the study, RSG and JJ wrote the manuscript, RSG analysed the data, JJ acquired funding and managed the project. All authors reviewed and edited the draft version.

*Competing interests.* The contact authors have stated that none of of the authors have conflicting interests

ther geographical representation in this paper. While Copernicus Publications makes every effort to include appropriate place names, the final responsibility lies with the authors.

*Acknowledgments.* The postdoc funding of R. Sosa-Gutierrez was supported by IRD. This study was supported by the projects ANR FORESEA (https://sargassum-foresea.cnrs.fr; ANR-19-SARG-0007) and TOSCA-SAREDA-HR and TOSCA-SARGAT funded by CNES. The *Sargassum* cover database was processed by AERIS/ICARE Data and Services Center at the University of Lille, in collaboration with the MIO (doi:10.12770/8fe1cdcb-f4ea-4c81-8543-50f0b39b4eca). Finally, we would like to thank the reviewers Maria Josefina Olascoaga and Clément Vic, their valuable comments helped improve this article.

*Open Research.* The remote sensing ADT observations are available on Global Ocean Gridded L 4 Sea Surface Heights And Derived Variables Reprocessed 1993 Ongoing | Copernicus Marine Service and the *Sargassum* fractional cover is available at https://www.odatis-ocean.fr.

*Review statement.* This paper was edited by Erik van Sebille and reviewed by Maria Josefina Olascoaga and Clément Vic.

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
