# Peer review of "Sargassum accumulation and transport by mesoscale eddies"

_EGUsphere, 2025_

## Referee Comment (RC1)

**Review of "*Sargassum* accumulation and transport by mesoscale eddies," by Sosa–Gutiérrez et al.**

This study investigates the impact of eddies on the distribution and movement of *Sargassum* in the tropical Atlantic. By utilizing 13 years of satellite data, it demonstrates that both cyclonic and anticyclonic eddies can trap and transport *Sargassum*. The study also reveals that *Sargassum* tends to accumulate and cover more area in cyclonic eddies, while it is less abundant and diminishes in anticyclonic eddies.

I appreciate the extensive efforts involved in analyzing such a large geographic area over an extended period. It is encouraging to see the theoretical results of Beron-Vera & Miron (2020) effectively tested and validated. However, I have some comments and suggestions that should be addressed before a final recommendation can be made.

1. My primary concern pertains to the methodology employed for detecting eddies. It is widely recognized that the method outlined in the paper is observer-dependent. The authors should incorporate a discussion addressing the potential issues associated with this methodology. For a detailed analysis on this subject, refer to Andrade-Canto, F., & Beron-Vera, F. J. (2022). "Do eddies connect the tropical Atlantic Ocean and the Gulf of Mexico?" *Geophysical Research Letters* 49, e2022GL099637.

2. In line 30, the citation Beron-Vera et al. (2021) should be corrected to Beron-Vera (2021). Additionally, the paragraph lacks clarity and should be rewritten.

3. Around line 40, it is inaccurately stated that Andrade et al. (2022) investigated only a single instance. In reality, an 8-year-long record was analyzed in the Caribbean Sea, where the study discovered *Sargassum* in both cyclonic and anticyclonic formations in similar quantities.

4. Vic et al. (2022) do not significantly contribute to the issue explored in this paper. Based on my understanding, Vic et al. used drogued drifters in their study. Employing undrogued drifters instead could potentially provide more valuable insights.

5. Examining Figure 3e raises questions about the methodology used for eddy detection, as there is no evident pattern in the tracer distribution to corroborate the presence of an eddy.

6. The authors observe a greater accumulation in cyclones. However, what is the ratio of cyclones to anticyclones in the region, regardless of *Sargassum* presence?

---

## Referee Comment (RC2)

Review of manuscript *Sargassum* **accumulation and transport by mesoscale eddies** by Rosemery Sosa-Gutierrez et al. (egusphere-2025-514.pdf)

Review by Clément Vic (LOPS, Plouzané, France) on 4[th] March 2025.

The authors investigate how sargassum is trapped and transported within mesoscale eddies in the tropical Atlantic. They use a combination of eddy identification and tracking based on satellite altimetry, and satellite-derived fractional cover of sargassum, in a composite approach. They find that mesoscale cyclonic eddies (CEs) contain on average 15% more sargassum than mesoscale anticyclonic eddies (AEs), which is consistent with the literature that reports such an asymmetry. Interestingly, the asymmetry seemingly grows along the eddy lifecycle. The methodology is robust and results are clearly exposed, although rapidly discussed. I recommend the manuscript for publication in Ocean Science. I only have a couple of minor comments that I hope can help to clarify some points. None of the comments fundamentally questions the methods or results.

**Minor comments**

1. The introduction misses some discussion and references to recent studies on the role of submesoscales in structuring floating material, near-surface boundary layer material, and ecosystems, e.g., D'Asaro et al. (2018); Lévy et al. (2018); Esposito et al. (2021). Submesoscales seem to be of paramount importance in clustering material in boundary layers.
2. L. 32, the reference to Provenzale (1999) is wrongly used. Quote from Provenzale (1999): "While in a nonrotating reference frame heavy impurities are always ejected from coherent vortices, in rotation-dominated systems, the Coriolis force may become stronger than the centrifugal term when the Rossby number is small. As a consequence, **heavy impurities can be concentrated in the cores of anticyclonic vortices** (Tanga et al 1996, Provenzale et al 1998)." See also their Figure 10.
3. L. 48-58: I find this paragraph ambiguous. What is the link between AFAI, MODIS and SAREDA? It is unclear to me if the authors performed any processing on MODIS "raw" images or if they used an on-the-shelf climatology of sargassum biomass / concentration.
4. L. 98: I am not sure I fully understand why the methodology would introduce a bias that would favour the detection of eddies with increased sargassum. It would be good to detail why and perhaps lean on references on composite approaches? I like that the authors address a potential methodological bias though!
5. Figure 3: I wonder if you could come up with a physical interpretation for the meridional shift between the peak of FC and the eddy centre for CEs in Figures 3b and 3f. Ekman pumping from a systematic wind stress difference on both sides of the eddies? …
6. Figure 5: I am confused by the colour code, contrasts are too light I think. To me, the dark red boxes are above light red boxes, but dark blue boxes are below light blue boxes. Also, I think it would be insightful to give the biomass outside of eddies, defined as the residual between total and within eddies (AEs and CEs, with an outer limit of say, between 1.5 and 3 radii?)
7. L. 134: Could you be more quantitative? What is the ratio of the surface covered by AEs vs CEs and how does it compare to the biomass within AEs vs CEs?
8. L. 142 and Figure 7: The fact that CEs increase their sargassum biomass within their lifetime is very interesting. This is different from the hypothesis made in Vic et al. (2022) that submesoscale processes prior to the mesoscale eddy formation are instrumental at gathering material. So, is there any mechanism that could help explain this result? I thought of Ekman pumping à la Gaube et al. (2015), considering that winds in the area are dominated by westward Trade winds… Also, I wonder if using a normalized time scale is physically meaningful. You would merge eddy characteristics at very different stages.

**Wording, typos, etc.**

- L. 10: was not … so far → has not been
- L. 50: remove "chain"
- L. 55: the tracking of high-frequency decay? → generation of near-inertial waves? Please clarify.
- L. 70: strongly → partially? I would not say that Figure 1 strongly supports this statement. It hints at a mesoscale influence, but there remain areas with high FC and no eddy and vice-versa.

- Figure 1: It would good to see the boxes for the three regions discussed in the manuscript. Also, in the caption, I would recall that only eddies with non-zero FC are shown.
- Figure 5: typo "Radio" in panels c,f,i
- Figure 6: Why is the colour range so different from previous plots? (x10^-5)
- Figure 8: Are red and blue histograms stacked or are they zero-based? Please clarify.
- L. 171: "connectors"? unclear, please rephrase.
- L. 184: Nutrient availability, and more generally, vertical motions, are driven at submesoscales more than mesoscales, e.g., Uchida et al. (2019); Picard et al. (2024).

**References**

D'Asaro, E. A., Shcherbina, A. Y., Klymak, J. M., Molemaker, J., Novelli, G., Guigand, C. M., et al. (2018). Ocean convergence and the dispersion of flotsam. *Proceedings of the National Academy of Sciences*, *115*(6), 1162–1167. https://doi.org/10.1073/pnas.1718453115

Esposito, G., Berta, M., Centurioni, L., Lodise, J., Ozgokmen, T., Poulain, P.-M., et al. (2021). Submesoscale vorticity and divergence in the Alboran Sea: Scale and depth dependence. *Frontiers in Marine Science*, *8*, 843. https://doi.org/10.3389/fmars.2021.678304

Gaube, P., Chelton, D. B., Samelson, R. M., Schlax, M. G., & O'Neill, L. W. (2015). Satellite observations of mesoscale eddy-induced Ekman pumping. *Journal of Physical Oceanography*, *45*(1), 104–132. https://doi.org/10.1175/JPO-D-14-0032.1

Lévy, M., Franks, P. J., & Smith, K. S. (2018). The role of submesoscale currents in structuring marine ecosystems. *Nature Communications*, *9*(1), 4758. https://doi.org/10.1038/s41467-018-07059-3

Picard, T., Gula, J., Vic, C., & Mémery, L. (2024). Seasonal tracer subduction in the Subpolar North Atlantic driven by submesoscale fronts. *Journal of Geophysical Research: Oceans*, *129*(9), e2023JC020782.

Provenzale, A. (1999). Transport by coherent barotropic vortices. *Annual review of fluid mechanics*, *31*(1), 55-93.

Uchida, T., Balwada, D., Abernathey, R., McKinley, G., Smith, S., & Levy, M. (2019). The contribution of submesoscale over mesoscale eddy iron transport in the open Southern Ocean. *Journal of Advances in Modeling Earth Systems*, *11*(12), 3934-3958.

Vic, C., Hascoët, S., Gula, J., Huck, T., & Maes, C. (2022). Oceanic mesoscale cyclones cluster surface Lagrangian material. *Geophysical Research Letters*, *49*, e2021GL097488. https://doi.org/10.1029/2021GL097488

---

## Author Comment (AC1)

*Author's reply to Reviewer 1:*

Authors response to comments by to the manuscript " Sargassum accumulation and transport by mesoscale eddies " by Sosa-Gutierrez et al. (rsosa@mercator-ocean.fr, julien.jouanno@ird.fr). We would like to thank Maria Josefina Olascoaga for the detailed and helpful comments to improve the manuscript. Below, we use black text for comments and green text for our response.

**1. My primary concern pertains to the methodology employed for detecting eddies. It is widely recognized that the method outlined in the paper is observer-dependent. The authors should incorporate a discussion addressing the potential issues associated with this methodology. For a detailed analysis on this subject, refer to Andrade-Canto, F., & Beron-Vera, F. J. (2022). \Do eddies connect the tropical Atlantic Ocean and the Gulf of Mexico?" Geophysical Research Letters 49, e2022GL099637.**

Thank you very much for raising this issue, we have now added the following sentence in the discussion section:

"*The eddy detection method used in this study (Chaigneau et al., 2009) has been widely applied in the literature (e.g., Pegliasco et al., 2021; Ernst et al., 2023). Like any method, however, it has certain limitations. Its performance can be influenced by the resolution of the satellite data and the specific detection criteria employed. It is also important to exercise caution when interpreting Eulerian mesoscale eddy detections, as these rely on streamline-based approaches that are inherently observer-dependent (Andrade and Beron-Vera, 2022). In contrast, Lagrangian methods—such as geodesic eddy detection—identify eddies as coherent material structures that resist stretching and deformation over time, and are therefore better suited to capturing flow-invariant transport pathways. In this work, we opted for an Eulerian detection method due to its relative ease of implementation. While it would be possible to reproduce this analysis using the methodology of Andrade and Beron-Vera (2022), we do not expect fundamentally different results, particularly because our detection method does not identify eddies that cross the Antilles Arc or the Yucatan Channel (see Figure 2), as was the case in Huang et al. (2021). That said, Lagrangian approaches would be particularly valuable for investigating transport and accumulation mechanisms, especially in scenarios where eddies re-form in the wake of island chains.*"

**2. In line 30, the citation Beron-Vera et al. (2021) should be corrected to Beron-Vera (2021). Additionally, the paragraph lacks clarity and should be rewritten.**

Thank you for your observation, the citation is now corrected and elsewhere. We rewrote the paragraph as follows, and we hope it has improved its clarity:

"*Sargassum remains afloat in the upper ocean due to its gas-filled bladders, making it highly responsive to both surface currents and wind. The dynamics of the upper ocean play a critical role in the formation of Sargassum accumulations, which can occur across a wide range of spatial scales (see Ody et al., 2019). At smaller scales, on the order of tens of meters, accumulations are typically driven by Langmuir circulation (Langmuir, 1938). At larger scales, reaching hundreds of kilometers, mesoscale and submesoscale frontal dynamics become dominant (Gower et al., 2013; Zhong et al., 2012). In particular, convergence zones associated with submesoscale dynamics have been shown to concentrate buoyant material and to structure the ecosystem (D'Asaro et al., 2018; Esposito et al., 2021, Lévy et al. 2018). However, the role of mesoscale eddies in the accumulation and transport of Sargassum remains uncertain. Early theoretical and experimental work by Provenzale (1999) suggested that only cyclonic eddies can retain floating particles. More recent advances, incorporating*"

*wind drag and elastic forces into the Maxey–Riley equations, have shown that these additional factors can have opposing influences—sometimes favoring accumulation in anticyclones rather than cyclones (Beron-Vera, 2021). Observational studies also present a mixed picture. For instance, limited in situ measurements have shown microplastic accumulation within anticyclonic eddies (Brach et al., 2018), while a more systematic study by Vic et al. (2022) demonstrated a tendency for drifters to accumulate in cyclonic structures in the North Atlantic. These contrasting findings highlight the complexity and context-dependence of floating object dynamics in mesoscale eddies. The accumulation and transport behavior likely depend on the specific properties of the objects in question—especially their buoyancy and windage.”*

**3. Around line 40, it is inaccurately stated that Andrade et al. (2022) investigated only a single instance. In reality, an 8-year-long record was analyzed in the Caribbean Sea, where the study discovered Sargassum in both cyclonic and anticyclonic formations in similar quantities.**

We apologize for the misquote, it is now corrected:
*“Andrade-Canto et al. (2022), using 8 years of satellite altimetry in the eastern Caribbean Sea, showed that mesoscale eddies (both cyclonic and anticyclonic) can carry out Sargassum transport.”*

**4. Vic et al. (2022) do not signicantly contribute to the issue explored in this paper. Based on my understanding, Vic et al. used drogued drifters in their study. Employing undrogued drifters instead could potentially provide more valuable insights.**

We agree that the undrogued drifter would probably better reflect the Sargassum dynamics, nevertheless their results show preferential accumulation in cyclonic eddies from their simulation using surface velocity, in line with our study. We now added in the text “drogued drifters”.

**5. Examining Figure 3e raises questions about the methodology used for eddy detection, as there is no evident pattern in the tracer distribution to corroborate the presence of an eddy.**

Our interpretation is different and given in the discussion:
“The central Atlantic region only had 1739 AE detected, against >4000 eddies for all other diagnostics (see Fig 5). In this region where mesoscale is weak, this sample size is probably too small for the average to converge.
*Results also suggest that the trapping is much more effective in the Caribbean than in the Central Atlantic, as revealed by sharpest contrast of Sargassum cover between the interior and exterior in the Caribbean. One hypothesis is that there is a much less energetic mesoscale activity in the Central Atlantic compared to the Caribbean: weak eddy circulation and the drift associated with the trade winds may reduce the capacity of the eddies to accumulate Sargassum in the Central Atlantic region.”*

**6. The authors observe a greater accumulation in cyclones. However, what is the ratio of cyclones to anticyclones in the region, regardless of Sargassum presence?**

Thank you for your proposition, we added a new table (Table 1) with ratio between the areas and biomass associated with the AEs and CEs, and discuss them:
“*To quantify these differences in Table 1 we show the ratio of total area and biomass ratio of CEs vs AEs. In the Caribbean, while the area ratio CEs/AEs is of 0.54, the accumulation of Sargassum in CEslead to a biomass ratio CEs/AEs of 0.80.*”

---

## Author Comment (AC2)

*Author's reply to Reviewer 2:*

Authors response to comments by to the manuscript " Sargassum accumulation and transport by mesoscale eddies " by Sosa-Gutierrez et al. (rsosa@mercator-ocean.fr, julien.jouanno@ird.fr). We would like to thank Clément Vic for the detailed and helpful comments to improve the manuscript. Below, we use black text for comments and green text for our response.

1. **The introduction misses some discussion and references to recent studies on the role of submesoscales in structuring floating material, near-surface boundary layer material, and ecosystems, e.g., D'Asaro et al. (2018); Lévy et al. (2018); Esposito et al. (2021).**

Thank you for your review and suggestion. We have expended the references in the introduction: "*In particular, convergence zones associated with submesoscale dynamics have been shown to concentrate buoyant material and to structure the ecosystem (D'Asaro et al., 2018; Esposito et al., 2021, Lévy et al. (2018)*"

2. **L. 32, the reference to Provenzale (1999) is wrongly used. Quote from Provenzale (1999): "While in a nonrotating reference frame heavy impurities are always ejected from coherent vortices, in rotation-dominated systems, the Coriolis force may become stronger than the centrifugal term when the Rossby number is small. As a consequence, heavy impurities can be concentrated in the cores of anticyclonic vortices (Tanga et al 1996, Provenzale et al 1998)." See also their Figure 10.**

We agree that we made a shortcut in the interpretation of Provenzale (1999). This is now clarified and expended as follows :
"*Early theoretical and experimental work by Provenzale (1999) suggested that heavy impurities can be concentrated in the cores of anticyclonic vortices. Beron-Vera et al. (2015) provided both theoretical justification and numerical evidence for a more general principle governing the behavior of inertial particles near quasigeostrophic eddies: anticyclonic (cyclonic) eddies tend to attract heavy (light) particles and repel light (heavy) ones, respectively.*"

3. **L. 48-58: I find this paragraph ambiguous. What is the link between AFAI, MODIS and SAREDA? It is unclear to me if the authors performed any processing on MODIS "raw" images or if they used an on-the-shelf climatology of sargassum biomass / concentration.**

Thanks, we agree it was not clear. The description has been rephrased as follows:
"*Sargassum detections were obtained from the SAREDA database (Sargassum Evolving Distributions in the Atlantic, Descloitres et al., 2021). This product estimates Sargassum cover by computing the Alternative Floating Algae Index (AFAI; Wang and Hu 2016) from ocean color acquisitions by the Moderate Resolution Imaging Spectroradiometer (MODIS), which operates aboard the Aqua and Terra satellites. The AFAI, computed using the processing chain described in Descloitres et al. (2021), was converted to Fractional Cover (FC), which represents the proportion of Sargassum cover in each pixel. Daily FC at 1 km from the SAREDA database were aggregated on a regular grid of 0.25° (~25 km) horizontal resolution.*"

4. **L. 98: I am not sure I fully understand why the methodology would introduce a bias that would favour the detection of eddies with increased sargassum. It would be good to detail why and perhaps lean on references on composite approaches? I like that the authors address a potential methodological bias though!**

We try to better explicit our point as follows :
*"Since the composites are constructed using eddies where Sargassum presence is detected at least once during the eddy's lifetime, this could introduce a bias favoring instances of Sargassum being trapped inside eddies. To verify that, we performed a null hypothesis test by compositing the Sargassum distribution with the same criteria but using the eddy contours for the previous year."*

5. **Figure 3: I wonder if you could come up with a physical interpretation for the meridional shift between the peak of FC and the eddy centre for CEs in Figures 3b and 3f. Ekman pumping from a systematic wind stress difference on both sides of the eddies?**

Thank you for this observation. We missed it. We add the following comment:
*"Also, note in Figure 3b a meridional offset between the location of the maximum FC peak and the center of the eddies—unlike the Caribbean composites (Figure 3d), where the Sargassum distribution appears centered within the eddies. The pattern observed in Figure 3b may result from the influence of Central Atlantic cases included in the composite, which show a southward displacement of Sargassum relative to the eddy centers (see Figure 3f). The cause of this southward shift is not fully understood at this stage, but it may be linked to windage effects from the trade winds or to the background distribution of Sargassum, which generally tends to accumulate along the Intertropical Convergence Zone (ITCZ). Figure 4 also shows this latitudinal gradient of FC (see 4b and 4d), which means that this gradient is a result of sampling this peculiar region with this eddy scale."*

6. **Figure 5: I am confused by the colour code, contrasts are too light I think. To me, the dark red boxes are above light red boxes, but dark blue boxes are below light blue boxes. Also, I think it would be insightful to give the biomass outside of eddies, defined as the residual between total and within eddies (AEs and CEs, with an outer limit of say, between 1.5 and 3 radii?)**

Thank you for your feedback regarding the color contrast. We modified the figure for better visualization. The biomass outside the eddies can already be inferred from Figure 8 so we prefer not to add it again here.

7. **L. 134: Could you be more quantitative? What is the ratio of the surface covered by AEs vs CEs and how does it compare to the biomass within AEs vs CEs?**

We have added a table where the ratio between the total areas of AEs and CEs is compiled and shown, as well as the ratio of the biomass of the eddies, and add the following text:
*"To quantify these differences in Table 1 we show the ratio of total area and biomass ratio of CEs vs AEs. In the Caribbean, while the area ratio CEs/AEs is of 0.54, the accumulation of Sargassum in CEs lead to a biomass ratio CEs/AEs of 0.80."*

8. **L. 142 and Figure 7: The fact that CEs increase their sargassum biomass within their lifetime is very interesting. This is different from the hypothesis made in Vic et al. (2022) that submesoscale processes prior to the mesoscale eddy formation are instrumental at gathering material. So, is there any mechanism that could help explain this result? I thought of Ekman pumping à la Gaube et al. (2015), considering that winds in the area are dominated by westward Trade winds… Also, I wonder if using a normalized time scale is physically meaningful. You would merge eddy characteristics at very different stages.**

We agree this is very interesting. We have no preferred hypothesis so far. We choose this methodology of normalized time scale as a first and basic attempt to see the evolution over

time. This could be refined. In some sense this result is coherent with Sargassum Lagrangian model from Beron-Vera (2021). Our next step will be to further understand this behavior and investigate whether other ingredients could play: windage, non-linear Ekman pumping, regional dynamics or air-sea mesoscale coupling effect.

**Wording, typos, etc.**

L. 10: was not … so far → has not been
Corrected

L. 50: remove "chain"
Corrected

- L. 55: the tracking of high-frequency decay? → generation of near-inertial waves? Please clarify.
Corrected as follows : *They have already allowed the tracking of rapid decrease in Sargassum coverage in the lee of tropical cyclones (Sosa-Gutierrez et al. 2022)*

L. 70: strongly → partially? I would not say that Figure 1 strongly supports this statement. It hints at a mesoscale influence, but there remain areas with high FC and no eddy and vice-versa. Corrected

Figure 1: It would good to see the boxes for the three regions discussed in the manuscript. Also, in the caption, I would recall that only eddies with non-zero FC are shown.
Thanks, we added the boxes and completed the caption.

Figure 5: typo "Radio" in panels c,f,i,
Corrected, thanks

Figure 6: Why is the colour range so different from previous plots? (x10^-5)
Thanks for the point this is. It was an error and it has been corrected.

Figure 8: Are red and blue histograms stacked or are they zero-based? Please clarify.
They are stacked and this was clarified, thanks.

L. 171: "connectors"? unclear, please rephrase
We agree and we removed the sentence.

L. 184: Nutrient availability, and more generally, vertical motions, are driven at submesoscales more than mesoscales, e.g., Uchida et al. (2019); Picard et al. (2024).
Indeed, but they are also evidence that cyclonic/anticyclonic mesoscale shape the nutrient availability (Gaube et al., 2014, McGillicuddy, 2016, Damien et al. 2021)

Damien, P., Sheinbaum, J., Pasqueron de Fommervault, O., Jouanno, J., Linacre, L., & Duteil, O. (2021). Do Loop Current eddies stimulate productivity in the Gulf of Mexico?. *Biogeosciences Discussions*, *2021*, 1-52.

---

## Referee Report (RR1)

Review of manuscript ***Sargassum* accumulation and transport by mesoscale eddies** by Rosemery Sosa-Gutierrez et al. (egusphere-2025-514-manuscript-version2.pdf)

Review by Clément Vic (LOPS, Plouzané, France) on 22nd April 2025.

This is my second review of the manuscript. The authors replied favourably to my comments and I recommend the manuscript for publication in Ocean Science. I just spotted a few very minor points; they are listed below. Line numbers are relative to the "track changes" document. I do not need to see a revised manuscript.

- L. 32: remove "the" before "ecosystems" (it should be plural)
- L. 53: can be transporters of *Sargassum* → can transport *Sargassum*
- L. 55: can carry out *Sargassum* transport → can transport *Sargassum*
- Section 1 ends with a list of eddy detection methods. I think it would be good to (1) close this paragraph in mentioning what method you opted for and why, and (2) have a short paragraph afterward to announce the plan of the article.
- L. 68: remove "chain"
- L. 137: no dot after Figure.
- L. 143: low → small
- L. 169: correspond → corresponds
- L. 181: it is worth noting that (no comma)
- L. 192: remove "the" before "AEs"
- L. 206 (and at other locations): move references to the end of the sentence (if possible, preserving their meaning)
- L. 232: "this sample size is probably too small for the average to converge." Could you back up this statement? maybe in subsampling other regions?
- L. 241: I find "overall growth" unclear. Are you refering to the total biomass ?

---

## Author Response (AR2)

*Author's reply to Reviewer 2:*

Authors response to comments by to the manuscript " Sargassum accumulation and transport by mesoscale eddies " by Sosa-Gutierrez et al. (rsosa@mercator-ocean.fr, julien.jouanno@ird.fr). We would like to thank Clément Vic for the detailed and helpful comments to improve the manuscript. Below, we use black text for comments and green text for our response.

L. 32: remove "the" before "ecosystems" (it should be plural)
Corrected

L. 53: can be transporters of Sargassum → can transport Sargassum
Corrected

L. 55: can carry out Sargassum transport → can transport Sargassum
Corrected

Section 1 ends with a list of eddy detection methods. I think it would be good to (1) close this paragraph in mentioning what method you opted for and why, and (2) have a short paragraph afterward to announce the plan of the article.

We add the following sentence : "*In this work, we opted for an Eulerian detection method due to its relative ease of implementation. This paper is organized as follows. Section 2 describes the data and methods used. The main characteristics of Sargassum cover in cyclonic and anticyclonic eddies are presented in the results section, section 3. Finally, in section 4, we present the summary and discussion of our results.*"

L. 68: remove "chain"
Corrected

L. 137: no dot after Figure.
Corrected

L. 143: low → small
Corrected

L. 169: correspond → corresponds
Corrected

L. 181: it is worth noting that (no comma)
Corrected

L. 192: remove "the" before "AEs"

Corrected

L. 206 (and at other locations): move references to the end of the sentence (if possible, preserving their meaning)

The text has been modified as follows : "*Using ADT satellite imagery, the distribution of Sargassum detections from 2011 to 2023 was analyzed, by applying an eddy detection and tracking methodology (Chaigneau et al., 2009; Pegliasco et al., 2015*)."

L. 232: "this sample size is probably too small for the average to converge." Could you back up this statement? maybe in subsampling other regions?

As we are not fully convinced of this sentence, we removed this sentence and the one before : *"The central Atlantic region only had 1739 AE detected, against >4000 eddies for all other diagnostics (see Fig 5). In this region where mesoscale is weak, this sample size is probably too small for the average to converge."*

We think the paragraph is much more clear. Thanks

L. 241: I find "overall growth" unclear. Are you refering to the total biomass ?

In addition to contrasted surface radial velocity properties between AEs and CEs, this could contribute to the contrasted time evolution of the distribution highlighted in Figure 7 and could modulate overall growth of total *Sargassum* biomass at the basin scale.